# ESBL-Producing Enterobacterales at the Human–Domestic Animal–Wildlife Interface: A One Health Approach to Antimicrobial Resistance in Piauí, Northeastern Brazil

**DOI:** 10.3390/vetsci11050195

**Published:** 2024-04-28

**Authors:** Sandy Kelly S. M. da Silva, Danny A. Fuentes-Castillo, Ana Carolina Ewbank, Carlos Sacristán, José L. Catão-Dias, Anaiá P. Sevá, Nilton Lincopan, Sharon L. Deem, Lauro C. S. Feitosa, Lilian S. Catenacci

**Affiliations:** 1Programa de Pós-Graduação Saúde Animal na Amazônia, Universidade Federal do Pará (UFPA), Belém 66075-110, Brazil; 2Departamento de Patología y Medicina Preventiva, Facultad de Ciencias Veterinarias, Universidad de Concepción (UDEC), Concepción 4070409, Chile; dannyfuentes@udec.cl; 3Centro de Investigación en Sanidad Animal (CISA-INIA), Consejo Superior de Investigaciones Científicas (CSIC), 28130 Valdeolmos, Spain; ewbank@inia.csic.es (A.C.E.);; 4Laboratório de Patologia Comparada de Animais Selvagens (LAPCOM), Faculdade de Medicina Veterinária e Zootecnia, Universidade de São Paulo (USP), São Paulo 05508-270, Brazil; 5Departamento de Ciências Agrárias e Ambientais, Universidade Estadual de Santa Cruz (UESC), Bahia 45662-900, Brazil; 6Laboratório de Resistência Bacteriana e Alternativas Terapêuticas, Instituto de Ciências Biomédicas, Universidade de São Paulo (USP), São Paulo 05508-220, Brazil; 7Saint Loius Zoo, Institute for Conservation Medicine, St. Louis, MO 63110, USA; 8Centro de Inteligência em Agravos Tropicais Emergentes e Negligenciados (CIATEN) e Centro de Ciências Agrárias (CCA), Universidade Federal do Piauí (UFPI), Piauí 64049-550, Brazil; lcsfeitosa@ufpi.edu.br

**Keywords:** antimicrobial, wildlife, conservation medicine, microorganisms, extended spectrum beta-lactamase, *Escherichia coli*, *Klebsiella pneumoniae*

## Abstract

**Simple Summary:**

The inappropriate use of antibiotics has favored the adaptation of bacteria resistant to these drugs and is a growing problem in the 21st century. It may affect not only the health of humans but also domestic and wild animals. In this study, we investigated the risk factors and the presence of one type of antibacterial resistance present in the feces of domestic animals and free-living birds in the State of Piauí, Brazil. A total of 59 samples of the 387 (15.2%) analyzed showed bacterial resistance. Resistant bacteria were found in free-living animals that had never been treated with any medication and in domestic animals raised for subsistence. We hypothesize that the lack of access to veterinary care and information regarding antimicrobial therapy, along with access to antimicrobials without medical prescription, favors the inadequate use of antimicrobials in Piauí and, thus, contamination of the environment. We recommend an educational outreach platform and the development of public health policies that support the responsible use of antimicrobials in Piauí and other Brazilian states.

**Abstract:**

The use, misuse, and overuse of antimicrobials is one of the main public health threats of the 21st century. We investigated the risk factor of the presence of extended-spectrum, cephalosporin-resistant Enterobacterales in feces of non-domestic and domestic birds and other domestic animals in Piauí State, northeast Brazil. We collected a total of 387 cloacal and rectal swab samples of free-living birds, domestic birds, and domestic mammals in five municipalities: Amarante, Água Branca, Lagoa Alegre, Parnaíba, and Teresina. A total of 59/387 (15.2%) of these samples harbored extended spectrum beta-lactamase (ESBL)-producing Enterobacterales. Using the MALDI-TOF technique, we identified fifty-seven samples as *Escherichia coli* and two samples as *Klebsiella pneumoniae*. Teresina and Parnaíba had the highest prevalence of animals with resistant bacteria (32.1% and 27.1%, respectively) and highest exposure risk factor (OR of 16.06 and 8.58, respectively, and *p* < 0.001 for all). Multidrug-resistant, ESBL-producing Enterobacterales were observed in 72.8% of the samples (43/59). For the free-living birds, the positive samples belonged to a great kiskadee (*Pitangus sulphuratus*) and a semipalmated sandpiper (*Calidris pusilla*) in migratory and resident species, respectively. For domestic animals, the swine samples showed the highest prevalence of antimicrobial resistance. The lack of access to veterinary care and information regarding antimicrobial therapy, along with the easy access to antimicrobials without medical prescription, favors the inadequate use of antimicrobials in Piauí.

## 1. Introduction

The use, misuse, and overuse of antimicrobials is recognized by the World Health Organization as one of the main public health threats of the 21st century, leading to the accelerated emergence of antimicrobial resistance (AMR) and consequent health and economic burdens [1,2]. Bacteria develop resistance to antimicrobials through a natural process of evolution, classified as intrinsic; acquired via selective pressure associated with the use of a particular antimicrobial, or via horizontal transfer of antimicrobial resistance genes between bacteria [3,4].

Bacteria from the *Enterobacterales* order such as *Klebsiella* spp., *Escherichia coli*, *Salmonella* spp., and others are part of the gastrointestinal microbiota of vertebrates. However, they are also considered important human pathogens for favoring the spread of AMR as reservoirs of antibiotic resistance genes [5].

Some potential impacts of anthropogenic actions, such as urbanization, deforestation, and wild animal trafficking, may favor the spread of resistant bacteria or their resistance determinants within wild microbial communities [6,7,8]. Furthermore, the maintenance and spread of antimicrobial resistant genes (ARGs) between domestic animals, humans, and free-living birds may occur through environmental contamination and selective pressure caused by antimicrobial residues and their metabolites [9,10].

Free-living animals may serve as sentinels of rural and urban environmental health through the presence of antimicrobial resistant bacteria in their feces [11]. In humans, direct contact with animals, or through consumption and manipulation of contaminated food, may lead to exposure to resistant bacteria [12].

Antimicrobial resistance can be considered a classic example of a One Health challenge [13,14]. In other words, ARGs are examples of the interconnection between people, other animals, plants, and the environment shared by all, since the misuse of many antimicrobials may create favorable conditions for bacteria to fix resistance genes that can be transmitted to humans and animals through contaminated food or the environment. Beyond the human–animal interface, the environment plays an important two-way role in the spread and maintenance of AMR, demonstrating the need for surveillance actions focusing on the One Health approach [15].

Although the Ministry of Health in Brazil created an AMR prevention and control plan based on the recommendations of the tripartite alliance (World Health Organization (WHO)/World Organization for Animal Health (WOAH)/Food and Agriculture Organization (FAO)) [16], studies are still scarce, especially in the north and northeastern regions of the country [17]. The Piauí State, located in northeastern Brazil, has several factors that favor the spread of AMR bacteria, including diverse rural communities adjacent to biodiversity rich areas, high poverty rates, and a population with little access to information and basic sanitation, along with the easy acquisition of antimicrobials over the counter [18,19,20,21].

In this study, we investigated the risk factors and the presence of extended spectrum beta-lactamase (ESBL)-producing Enterobacterales in the feces of domestic animals and free-living birds in Piauí State and provide recommendations for educational outreach and policy development that support the appropriate use of antimicrobials in Brazil.

## 2. Materials and Method

### 2.1. Study Area

This study was conducted in five municipalities of Piauí State, northeastern Brazil: Amarante, Água Branca, Lagoa Alegre, Parnaíba, and Teresina (Figure 1). We selected the municipalities in the northern region of the state based on the higher population density of humans and domestic animals than other areas of Piaui. In addition, the northern region is the main entry point for migratory birds in Piauí across the coast [22]. Piaui is characterized by the Cerrado and Caatinga biomes and has a climate characterized as semi-arid, tropical, hot, and humid, with an average temperature of 27 °C and humidity of 65% [23].

### 2.2. Study Period and Group

The project campaigns were carried out between October 2019 to October 2020. For all cardinal points, we performed convenience sampling of 15 samples from each animal group: non-domestic birds, domestic birds (chickens and ducks), horses, pigs, and ruminants (sheep, goats, and cattle). Sampling was conducted according to [24].

For each animal sampled, we recorded the date, time, location, geographic coordinates (using a Garmin GPSMAP 65s), species identification, sex (when possible), and estimated age. Additionally, for domestic animals, the clinical history and environmental conditions were collected, when available.

### 2.3. Bird and Domestic Sampling

To capture the free-living birds, we set eight mist nets (3 × 15 m with 30 mm) from 05:30 am to 9:30 am and again from 04:00 pm to 06:00 pm, during a period of 5–8 days per municipality. Once trapped in the net, each bird was removed and manually restrained. All birds were immediately released after sampling. The swabs were maintained in Amies culture medium with charcoal (TRANS-BAC AMIES DME) or Stuart (TRANS-BAC STUART DME) media, sealed and labeled, and kept at room temperature until processed at the Laboratory of Bacterial Resistance and Therapeutic Alternatives at the University of São Paulo (ICB-USP), São Paulo state, Brazil.

In each municipality, samples were collected at cardinal points (north, east, south, and west) to obtain distributed information from each location. We defined each location according to the forest remnants likely to capturing free-living birds and proximity to human populations and domestic animals.

Before collecting the samples from domestic animals, a survey questionnaire was carried out with the farmers to collect information regarding individual animals, including age, sex, history of recent illnesses, use of medications, housing, and purpose (i.e., consumption, labor, or company). The domestic animals were manually restrained, and the cloacal or rectal samples were obtained directly from the rectum by quickly inserting the swab, followed by rotational movements. Subsequently, all swabs were stored and labeled as described above. A Free and Clarified Consent Term was signed by all animal owners.

### 2.4. Laboratory Analysis

#### 2.4.1. Antimicrobial Susceptibility Testing and Molecular Analysis

In order to detect ESBL-producing Enterobacterales, swab samples were first inoculated on bacteriological culture medium MacConkey agar (MAC) (Oxoid, UK) supplemented with 2 µg/mL of ceftriaxone (CRO) and incubated overnight at 35 ± 2 °C [25].

Antimicrobial susceptibility testing was performed via the disc diffusion method using human and veterinary antibiotics according to Clinical and Laboratory Standards Institute (CLSI) [26,27] protocols, which included β-lactam + β-lactamase inhibitor combination (amoxicillin + clavulanic acid—AMC), cephems (ceftriaxone—CRO, cefotaxime—CTX, ceftiofur—CTF, and cefepime—CPM), monobactams (aztreonam—ATM), carbapenems (ertapenem—ETP, imipenem—IPM, and meropenem—MER), fluoroquinolones (enrofloxacin—ENO and ciprofloxacin—CIP), sulfonamides (sulfamethoxazole/trimethoprim—SUT), aminoglycosides (gentamicin—GEN and amikacin—AMI), tetracyclines (tetracycline—TET), and glycylcycline (tigecycline—TIG). *Escherichia coli* ATCC 25922 was used as a control strain. ESBL production was screened using the double-disc synergy test [28] (Table 1). Bacteria presenting resistance to more than two different classes of antimicrobials were classified as a multidrug-resistant bacteria [29].

From this stage, all samples that showed phenotypic characteristics of ESBL production were submitted to molecular tests, with DNA extraction according to the boiling method [30] followed by a PCR protocol using CTX-M, CTX-M-8, and CTX-M-15 primers and samples that demonstrated sensitivity to cephalosporins in the antibiogram without ESBL characteristics were submitted to molecular tests using the cephalosporinases primers SHV-1 and TEM-1 (Table A1) [31,32,33,34,35].

#### 2.4.2. Bacterial Identification

The ESBL-positive samples were individually streaked onto TSA agar plates and incubated overnight at 35 ± 2 °C. Morphologically distinct bacterial isolates were individually collected and identified via MALDI-TOF MS (Matrix Assisted Laser Desorption Ionization Time of Flight Mass Spectrometry, Bruker Daltonik, Leipzig, Germany) [36].

## 3. Ethics Statement

This study was performed in full compliance with the Biodiversity Information and Authorization System (SISBIO 63225-1) of the Brazilian Ministry of Environment, the National System for the Management of Genetic Heritage and Associated Traditional Knowledge (SISGEN A885B86), and the Ethical Committee in the Use of Animals from Federal University of Piauí (CEUA-UFPI process 646/2020).

## 4. Statistical Analysis

We used a multivariate linear regression model to compare the associated factors, considering the dependent variable as positive or negative for ESBL and independent variables as sex, age group, municipality, species group, and whether the animal’s purpose was food production. To select the best model, a backward approach was used, including variables that were significantly associated (*p* < 0.05) and had a minimum value of the Akaike Information Criterion (AIC). All these analyses were performed in the R Program with the package EpiDisplay. Significant variables were those with *p* < 0.05 and the 95% confidence interval was used for calculations. Univariate analyses were performed for variables that had less than 75% of the response into the formulary.

To compare the prevalence of resistance between the antibiograms for each individual species, we performed the Cochran Q test followed by the post hoc McNemar test, since the individuals are repeated (dependent samples), with Bonferroni correction for p value. To compare the prevalence of resistant strains between municipalities and between species, Student’s t test or the Chi-square test was used, with a Chi-square post hoc test and with Bonferroni correction for p value. For both analyses, we considered significance to occur when the p value was <0.05; all analyses were performed in the R program, using the rcompanion package.

## 5. Results

We collected a total of 387 cloacal and rectal swab samples, which included free-living wild birds of 43 different species (*n* = 89), domestic birds (chickens (*n* = 59) and ducks (*n* = 1)), and domestic mammals (horses (*n* = 80), pigs (*n* = 76), sheep (*n* = 8), goats (*n* = 71), and cattle (*n* = 3)) in five municipalities: Amarante (*n* = 85), Água Branca (*n* = 79), Lagoa Alegre (*n* = 75), Parnaíba (*n* = 70), and Teresina (*n* = 78) (Table 2). After laboratory analysis, the presence of ESBL-producing bacteria was found in 59 of these samples, totaling 15.2% of animals positive for this type of bacterial resistance. Using the MALDI-TOF technique we identified fifty-seven samples as *Escherichia coli* and two samples as *Klebsiella pneumoniae*. Among wild birds, three (3.37%) of them presented ESBL-producing Enterobacterales: great kiskadee (*Pitangus sulphuratus*), a common Neotropical bird, in Água Branca and two specimens of semipalmated sandpiper (*Calidris pusilla*), a migratory bird considered Near Threatened by the IUCN Red List, captured in Parnaíba.

Among the ESBL-positive samples, 76.2% (45/59) presented CTX-M beta-lactamase. From the samples with CTX-M, 10.1% (6/59) were CTX-M-15 and 6.7% (4/59) were CTX-M-8. Only one sample (1.6%, 1/59) was positive for SHV-1 and 1.6% (1/59) were positive for TEM-1. A total of 3.3% (2/59) did not have the gene identified through the primers used.

Teresina and Parnaíba had the highest prevalence of animals with resistant bacteria (32.1% and 27.1%, respectively) and the highest exposure risk factor (OR of 16.06 and 8.58, respectively, and *p* < 0.001 for all) (Table 3), in comparison to the Amarante municipality, with 5.9% of prevalence. The other municipalities had a prevalence lower than 10% (Table 3).

The animal group and municipality variables were selected using the multivariable model. Domestic birds, ruminants, and swine were the animal groups with the highest risk of being positive for ESBL-producing Enterobacterales (OR = 8.63, 11.66 and 39.68, respectively, and *p* < 0.01) (Table 3).

In univariate analyses, being male represented a protective factor for the presence of resistant bacteria (OR= 0.44, *p* = 0.001), with 10.8% of males being positive in contrast with 21.6% of positive females. Conversely, the prevalence of ESBL-producing Enterobacterales in young animals (less than one year old) was 30.4% significantly higher than in other age groups (OR= 2.72, *p* = 0.003) (Table 3).

Based on farmer surveys, only 32.1% (96/298) confirmed having used medications on their animals to treat or prevent any disease. From these 96 farmers, 25.0% (24/96) stated that the medications used were antimicrobials. However, none of the farmers recognized the name of the antimicrobials and had difficulties in answering this question. The majority (81.2%; 78/96) claimed to use only dewormers. However, in one of the interviews, for example, the “dewormer” used was noted by the researchers to be an antimicrobial from the tetracycline group. Ruminants were the animal group with the most confirmation of drug use, with 42.7% (41/96).

Considering the antibiogram results, 46.8% (200/427 antibiotic disks) of the samples were resistant to at least one class of antimicrobials (Figure 2). The sampled bacteria proved to be more resistant to antimicrobials from the beta-lactamases plus inhibitor (AMC), aminoglycosides (GEN and AMI), and carbapenem (ETP, IPM, and MER) groups and showed greater sensitivity to antimicrobials from the cephem (CRO, CTX, CTF, and CPM) group, a sub-group of β-lactam antimicrobials including cephalosporins and cephamycins (Figure 2).

Although the municipalities of Teresina and Parnaíba had the highest bacterial growth in the screening for extended-spectrum cephalosporin resistance than the other municipalities, there was no significant difference between sensitivity to each antimicrobial among the municipalities (X^2^ = 1.35, df = 4, *p*-value = 0.851) (Figure 3). Among the samples in each municipality, the cities located in the center of Piauí, Lagoa Alegre and Teresina, had the highest number of ESBL-producing Enterobacterales resistant to the aminoglycosides group (11.8%; 2/17 and 9.4%; 8/85, respectively) (Figure 3).

The Água Branca municipality had fewer animals with resistance to the monobactams group (4.8%; 1/21), compared to the other municipalities that had prevalences between 12.9% and 22.2%, but more animals resistant to tetracyclines (28.6%; 6/21), compared to the other municipalities that had prevalences between 11.1% and 17.6%.

Multi-resistant Enterobacterales were observed in 72.8% (43/59) of the samples (Figure 3). Among the ESBL-positive samples, 88.1% (52/59), 85.5% (50/59), and 79.6% (47/59) had resistance to the sulfonamides class, tetracyclines class, and monobactams class, respectively.

Despite the wide range in prevalence among species, from 35.7% for semipalmated sandpiper (*Calidris pusilla*) to 85.7% for cattle (*Bos taurus*), there was no significant difference between all of them (Fisher test, *p* = 0.399). However, it is noteworthy that domestic cattle (*Bos taurus*), ducks (*Anas platyrhynchos domesticus*), horses (*Equus ferus caballus*), and goats (*Capra aegragus hircus*) presented with more antimicrobial resistance bacteria than with sensitive ones. However, due to the low number and prevalence of individuals sampled, the confidence intervals did not allow for this hypothesis to be defended (Figure 4). An exception was cattle, whose prevalence was 85.7% and the lower confidence interval was less than 50% (95%; CI: 59.8%–100.0%), representing that for this species there are more resistant individuals to antimicrobials than susceptible.

## 6. Discussion

To the authors’ knowledge, this is the first study to detect beta-lactamase resistance genes in samples of Enterobacterales isolated from animals in Piauí. Beta-lactam antimicrobials are among the most frequently prescribed antimicrobials worldwide [37]. The massive use of extended-spectrum cephalosporins promotes selective pressure followed by a fast emergence of new beta-lactamases, which are able to degrade and confer resistance to these drugs [38,39].

The occurrence of ESBL may be underestimated globally, mainly due to the limitations of the laboratory tests (I), the differences between the detection and interpretation methods used in each country (II), and the underreporting of this phenomenon (as we saw in our research) (III), which all may lead to limiting the appropriate treatment of infections in both humans and animals [40].

In Brazil, studies on the presence of ESBL in domestic animals and free-living birds are still more scarce and were mainly concentrated in the southern and southeastern regions [41,42,43,44,45,46,47,48], with only a few studies conducted in the northeast [17,49,50]. The prevalence of ESBL-producing bacteria observed in these studies is considered variable due to the lack of a pattern in the number of samples, varying, for example, from 32 to 257 samples [50,51]. The elevated prevalence of ESBL-producing bacteria in Piauí should serve as a warning and provide the data necessary to develop awareness actions aimed at an educational approach and the development of public health policies to support the responsible use of antimicrobials in Piauí.

The CTX-M-ases constitute a rapidly growing cluster of enzymes, whose coding genes have worldwide dissemination and confer resistance to some antimicrobials, such as penicillin, extended-spectrum cephalosporins, and monobactams, which are inhibited by clavulanate, sulbactam, and tazobactam [52]. The CTX-M variants were found mainly in the southeastern region of Brazil among samples collected from humans in hospitals, most commonly associated with the presence of the *bla*CTX-M gene in *Klebsiella pneumoniae* and *E. coli* [53]. In this study, the high prevalence of cefotaximase (CTX-M-ase) genes found in samples of wild and domestic animals suggests the spread of these clinically relevant enzymes at least in Cerrado, Mangroves, and Caatinga biomes. These results suggest that more studies in different municipalities should be carried out to elucidate the panorama of bacterial resistance and the inclusion of human samples. We also suggest the sequencing of the positive samples from the present study to elucidate the genomic relationship, resistome, and virulome of these ESBL-producing Enterobacterales among the wild and domestic animal species.

Composed of a high number of rural communities that live under subsistence agriculture, Piauí presents a human population with limited access to basic sanitation, drinking water, and medical and veterinary assistance, with one of the worst Human Development Index (HDI) states in Brazil [54]. The lack of access to information and veterinary care, in addition to the easy acquisition of antimicrobials without a prescription, favors the use, misuse, and overuse of antimicrobials in this state, as observed in this study. The difficulty of obtaining information about medications offered to raise animals, confirms that the owners have access to drugs without knowing the purpose, appropriate dose, and correct protocol of use, as we saw in the farmer surveys, where the owners administered antimicrobials thinking they were dewormers. In spite of this, the higher prevalence of antimicrobial-resistant bacteria in swine without any treatment history highlights the already-present environmental contamination in the surveyed municipalities. Most of these animals were kept in inadequate environments, with poor hygiene and fed with leftover food, which can increase the chances of AMR development. Nevertheless, we recommend additional specific studies to test this hypothesis. The untreated sewage released into water bodies poses a considerably larger risk of spreading resistance genes, particularly considering that it is often used for irrigation of farmland and/or recreational human activities [55]. We also suggest that the water quality of the municipalities in this study should be investigated.

According to [56], one of the main barriers for resistant pathogen dispersal is having a proper hygiene routine. The municipalities of Teresina and Parnaíba, with some of the largest human populations (814.230,00 and 145.705,00 in habitants, respectively) in Piauí, also have the overall poorest sanitation levels (61.6% in Teresina and 23.5% in Parnaíba) [54]. This could explain the highest risk factor for animals living in these municipalities. These conditions added to the raising of domestic animals in inappropriate environments and may lead to contamination with antimicrobial resistant bacteria.

As expected, our findings revealed a higher AMR level in domestic animals in comparison to wild animals, suggesting that the proximity of domestic animals to humans may influence potential acquisition of resistant bacteria [57].

In this study, female animals and young animals presented a higher prevalence of resistant bacteria, which may be related to the use of antimicrobials during pregnancy as prophylaxis, to treat labor and udder problems during milk production in females, and as animal growth promoters in juveniles. Even though the Ministry of Agriculture and Livestock and Supply (MAPA) has vetoed the use of several of these substances in recent years, the use of antimicrobial agents for these purposes is not prohibited or controlled in Brazil [58].

ESBL-producing Enterobacterales were found in one sample from a great kiskadee and in one sample of semipalmated sandpiper, migratory and resident species, respectively. The presence of antimicrobial resistance has already been reported 900 km away from Piaui, in 32 individuals of semipalmated sandpiper by Silva, C. R. [49] in Paraíba state, although the tests used were unable to identify the resistant genes. To the authors’ knowledge, this is the first ESBL bacteria description in great kiskadee, a synanthropic bird commonly found in Brazil.

All positive wild bird samples were resistant to at least two antimicrobial groups, both from the cephem class. In Brazil, few studies have assessed the presence of ESBLs in wild birds [49,50,51,59,60]. The ESBLs were detected in one non-migratory species, magnificent frigatebird (*Fregata magnificens*), and four migratory species, semipalmated plover (*Charadrius semipalmatus*), ruddy turnstone (*Arenaria interpres*), semipalmated sandpiper, and lesser yellowlegs (*Tringa flavipes*)*,* all collected in the northeast of Brazil [49,50].

Our results corroborate the findings of [14], revealing that wild birds can carry strains of *E. coli* that are resistant to drugs of critical importance in human and veterinary medicine, suggesting that wild birds may serve as a reservoir for multidrug-resistant bacteria and resistance genes. However, these authors analyzed samples from pigeons and gulls, animals that have greater contact with humans and higher adaptation to anthropic areas, which may lead to increased contamination and a higher occurrence of antimicrobial resistant bacteria than in the samples from our study.

According to Gralha [61], antimicrobials from the β-lactam class are normally used as the drugs of first choice in infections caused by enterobacteria, due to their low toxicity and therapeutic efficacy. Although free-living birds have a low prevalence of ESBL-producing Enterobacterales present in their feces (3.4%), the finding of any ARG demonstrates the circulation and presence of these bacteria in the environment, as these wild birds are free-living and have never been treated with antimicrobials. The higher prevalence of antimicrobial resistance in non-domestic birds sampled in an anthropized place rather than in pristine areas was also observed in the study of [50] in Pernambuco, another state in the northeast region of Brazil.

As we found in the present study, multidrug-resistant *E. coli* has been frequently reported in farm animals in Brazil, as demonstrated in the review by [17]. Considering the screening method used, it was previously expected that all samples tested would be resistant to at least one class of antimicrobials. However, the high prevalence of multi-resistant strains (72.8%) shows a relevant phenomenon, since this may cause the limitation of viable antimicrobials for the treatment of diseases [62]. The sensitivity to aminoglycosides, including carbapenems found in our samples, confirms the data found by Dhanji, H. et al.; Abhilash, K.P.P. et al.; and Mezhoud, H. et al. [63,64,65,66]. As carbapenems are not hydrolyzed by ESBLs, its use is indicated in cases of ESBL-producing E. coli or K. pneumoniae infections, and ertapenem therapy was found to be more effective than imipenem or meropenem therapy in humans [67], which may indicate a good option for treatment in cases of ESBL infections in animals. The high prevalence of resistance in the samples from Piauí to sulfonamides (60.7%), tetracyclines (55.7%), and monobactams (52.5%) may be related to the inappropriate use of these drugs and the lack of information demonstrated by the owners who often treat their animals without veterinarian assistance. Most of the farmers reported medicating their animals without veterinary prescription and were often unable to say which drugs and doses they administered.

Due to their mechanism of action, low price, and availability, tetracyclines are one of the most commonly used antimicrobials in the world, and tetracycline-resistant strains have been widely identified to affect both humans and animals [68]. The same condition is observed regarding sulfonamides, commonly used in cases of diarrhea in domestic animals, thus favoring the emergence of resistant strains to this drug class [69]. Developing awareness programs on the correct use of antimicrobials is a crucial mechanism to reduce the rapid spread of ARGs in the State of Piauí.

## 7. Conclusions

The intersection of the human, animal, and ecosystem health—One Health—interface is recognized as being of key importance in the evolution and spread of antimicrobial resistance (AMR) and represents an important and yet rarely appreciated opportunity to undertake vital AMR surveillance.

The bacterial samples present in this study showed a high prevalence of antimicrobial resistance in the municipalities from Piauí. The inappropriate use of antimicrobials and a lack of proper drug class information identified in the farmers interviewed may be the main causes of resistant bacteria in this study. Brazil has few legislative means to control and monitor antimicrobial use. So, the implementation of programs for prevention and control of resistant bacteria or implementation of genomic AMR surveillance systems are even more necessary. Where systems are not yet in place, or to support harmonization of approaches and comparability of data globally, participation should be encouraged in the sentinel organism-focused WHO Tricycle protocol. This protocol aims for integrated global surveillance of one indicator, ESBL-producing *E coli*, across the human, animal, and environment sectors.

We also suggest the development of public health policies for the responsible use of antimicrobials in all animals and humans to improve the health of the environment, humans, and animals. We recommend that Piaui Animal Health agencies promote campaigns that show the importance of the correct use of antimicrobials and help limit the purchase of these medicines to doctor or veterinarian prescription.

Additionally, our results highlight the need for improvements in Piauí’s basic sanitation and proper collection and disposal of garbage to reduce environmental contamination and the proliferation of bacteria. Further research on antimicrobial resistance in the northeastern and northern regions is needed to better understand the scenario in Brazil.

## Figures and Tables

**Figure 1 vetsci-11-00195-f001:**
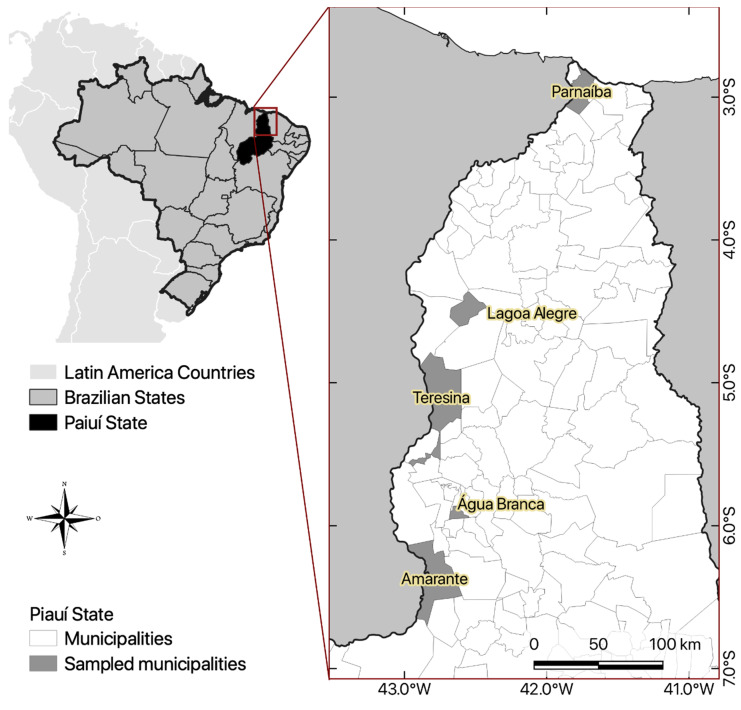
Study sample collection sites of ARGs in five municipalities of Piaui State, Brazil.

**Figure 2 vetsci-11-00195-f002:**
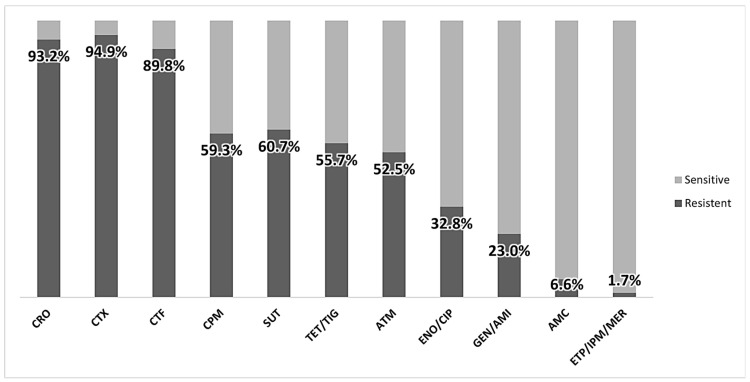
Groups of antimicrobials with the highest and lowest prevalence of resistance in extended-spectrum, cephalosporin-resistant Enterobacterales samples tested with different antimicrobials classes. AMC: Amoxicillin + clavulanic acid; CRO: Ceftriaxone; CTX: Cefotaxime; CTF: Ceftiofur; CPM: Cefepime; ATM: Aztreonam; ETP: Ertapenem; IPM: Imipenem; MER: Meropenem; ENO: Enrofloxacin; CIP: Ciprofloxacin; SUT: Sulfamethoxazole/trimethoprim; GEN: Gentamicin; AMI: Amikacin; TET: Tetracycline; TIG: Tigecycline.

**Figure 3 vetsci-11-00195-f003:**
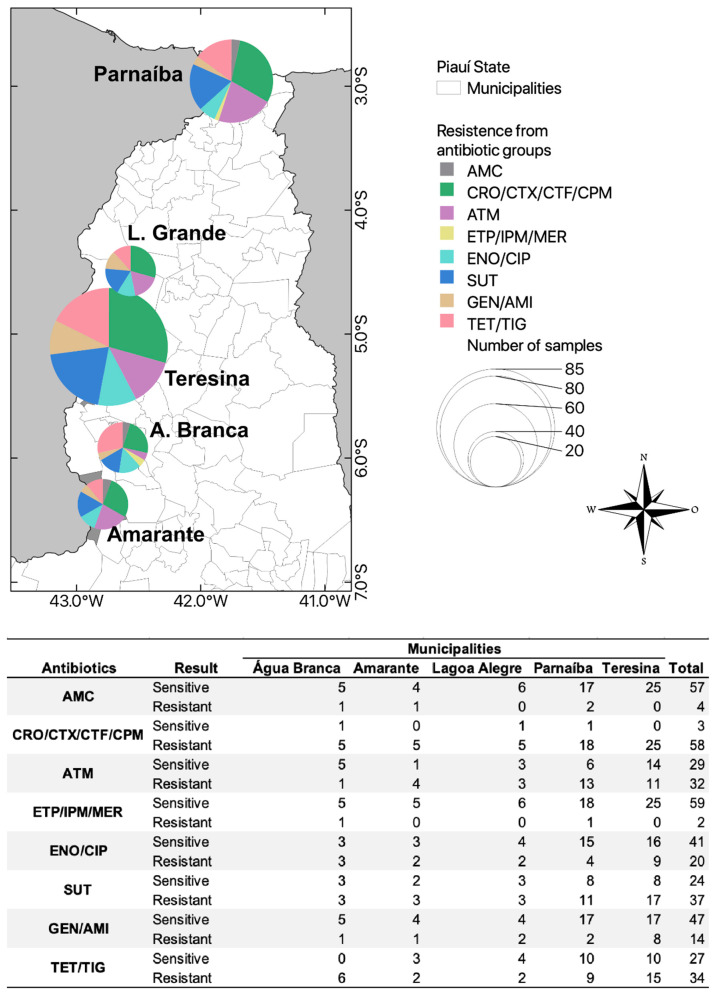
Prevalence of resistance to different classes of antimicrobials in extended-spectrum, cephalosporin-resistant Enterobacterales isolated from animals of five municipalities sampled north of Piauí State, Brazil. AMC: Amoxicillin + clavulanic acid; CRO: Ceftriaxone; CTX: Cefotaxime; CTF: Ceftiofur; CPM: Cefepime; ATM: Aztreonam; ETP: Ertapenem; IPM: Imipenem; MER: Meropenem; ENO: Enrofloxacin; CIP: Ciprofloxacin; SUT: Sulfamethoxazole/trimethoprim; GEN: Gentamicin; AMI: Amikacin; TET: Tetracycline; TIG: Tigecycline.

**Figure 4 vetsci-11-00195-f004:**
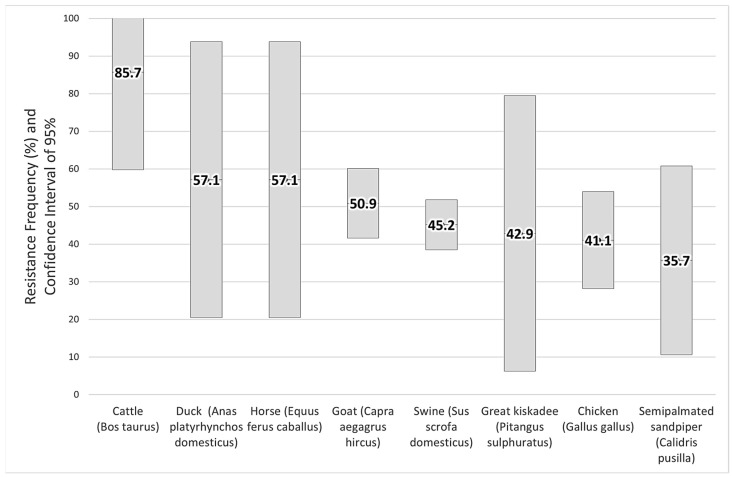
Prevalence of ESBL-producing Enterobacterales among species sampled, Piaui, Brazil.

**Table 1 vetsci-11-00195-t001:** Antibiotics used in antimicrobial susceptibility testing for detection of resistance genes in samples from Piauí, Brazil.

Classes	Antibiotics	Abbreviation	Concentration
β-lactam + β-lactamase Inhibitor Combination	Amoxicillin + clavulanic acid	AMC	30 μg
Cephems	Ceftriaxone	CRO	30 μg
	Cefotaxime	CTX	30 μg
	Ceftiofur	CTF	30 μg
	Cefepime	CPM	30 μg
Monobactams	Aztreonam	ATM	30 μg
Carbapenems	Ertapenem	ETP	10 μg
	Imipenem	IPM	10 μg
	Meropenem	MER	10 μg
Fluoroquinolones	Enrofloxacin	ENO	5 μg
	Ciprofloxacin	CIP	5 μg
Sulfonamides	Sulfamethoxazole/ trimethoprim	SUT	25 μg
Aminoglycosides	Gentamicin	GEN	10 μg
	Amikacin	AMI	30 μg
Tetracyclines	Tetracycline	TET	30 μg
Glycylcycline	Tigecycline	TIG	15 μg

**Table 2 vetsci-11-00195-t002:** Prevalence of ESBL-producing bacteria isolated from cloacal and rectal swab samples of domestic animals and free-living birds collected in five municipalities of Piauí State, Brazil.

Municipalities	Animal Group	n° of Samples	Positive Samples (*n*)	Resistance Genes	Prevalence/ Municipality (%)	Confidence Interval (95%)
Água Branca	Wild birds	17	1	blaSHV-1	5.88	0–17.07
	Domestic birds	13	1	blaCTX-M	7.69	0–22.18
	Equine	18	0	-	0	0
	Swine	16	2	blaCTX-M (1) Not identified (1)	12.5	0–28.71
	Ruminants	15	1	blaCTX-M-15	6.67	0–19.29
	Total	79	5	-	6.33	0.96–11.70
Amarante	Wild birds	18	0	-	0	0
	Domestic birds	12	3	blaCTX-M (3)	25.0	0.50–49.50
	Equine	16	0	-	0	0
	Swine	16	2	blaCTX-M (2)	12.5	0–28.71
	Ruminants	23	0	-	0	0
	Total	85	5	-	5.88	0.88–10.88
Lagoa Alegre	Wild birds	15	0	-	0	0
	Domestic birds	15	1	blaCTX-M-8	6.67	0–19.29
	Equine	15	1	blaCTX-M	6.67	0–19.29
	Swine	15	1	blaCTX-M	6.67	0–19.29
	Ruminants	15	2	blaCTX-M (2)	13.33	0–30.54
	Total	75	5	-	6.67	1.02–12.31
Parnaíba	Wild birds	14	2	blaCTX-M (2)	14.29	0–32.62
	Domestic birds	12	4	blaCTX-M (2)	33.33	6.66–60.01
	Equine	15	0	-	0	0
	Swine	14	11	blaCTX-M (9) blaCTX-M-15 (2)	78.57	57.08–100
	Ruminants	15	2	blaCTX-M (1) blaCTX-M-15 (1)	13.33	0–30.54
	Total	70	19		27.14	16.73–37.56
Teresina	Wild birds	25	0	-	0	0
	Domestic birds	8	0	-	0	0
	Equine	16	0	-	0	0
	Swine	15	14	blaCTX-M (12) blaCTX-M-8 (1) blaTEM-1 (1)	93.33	80.71–100
	Ruminants	14	11	blaCTX-M (7) blaCTX-M-8 (2) blaCTX-M-15 (1) Not identified (1)	78.57	57.08–100
	Total	78	25	-	32.05	21.69–42.41
All municipalities	Total	387	59	-	15.25	11.66–18.83

**Table 3 vetsci-11-00195-t003:** Univariable and multivariable relationships regarding the prevalence of ESBL-positive samples between different groups of analyses from fecal samples of domestic animals and free-living birds collected in five municipalities of Piauí State, Brazil.

		ESBL Positive		ESBL Negative					IC 95%		
		n	%	n	%	Total	Analyze	OR	Lower	Higher	*p* Value
Sex	Male	14	10.8	116	89.2	130	univariate	0.44	0.23	0.84	0.001
	Female	41	21.6	149	78.4	190		Ref			
	No Information	4	6.0	63	94.0	67		-			
Age Group	Cub	2	6.9	27	93.1	29	univariate	0.46	0.11	2.02	0.306
	Young	17	30.4	39	69.6	56		2.72	1.41	5.27	0.003
	Adult	40	13.8	250	86.2	290		Ref			
	Geriatric	0	0.0	3	100.0	3		0	0	Inf	0.987
	No Information	0	0.0	9	100.0	9		-			
Municipality	Água Branca	5	6.3	74	93.7	79	multivariate	1.11	0.29	4.24	0.874
	Amarante	5	5.9	80	94.1	85		Ref			
	Lagoa Alegre	5	6.7	70	93.3	75		1.15	0.30	4.36	0.842
	Parnaíba	19	27.1	51	72.9	70		8.58	2.75	26.80	<0.001
	Teresina	25	32.1	53	67.9	78		16.06	5.07	50.85	<0.001
Animal Group	Wild Birds	3	3.4	86	96.6	89	multivariate	ref			
	Domestic Birds	9	15.0	51	85.0	60		8.63	2.06	36.18	0.003
	Equine	1	1.3	79	98.8	80		0.42	0.04	4.22	0.460
	Ruminants	16	19.5	66	80.5	82		11.66	3.03	44.86	0.000
	Swine	30	39.5	46	60.5	76		39.68	10.26	153.54	0.000

## Data Availability

Data available on request from the authors. The data that support the findings of this study are available from the corresponding author, S. K. S.M. S., upon reasonable request.

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
