# Peer review of "ESBL-Producing Enterobacterales at the Human–Domestic Animal–Wildlife Interface: A One Health Approach to Antimicrobial Resistance in Piauí, Northeastern Brazil"

_vetsci, 2024, doi:10.3390/vetsci11050195_

Round 1

Reviewer 1 Report

Comments and Suggestions for Authors

The work is relevant and very important for epidemiological data regarding the presence of ESBL in Brazil.

The work is well written, the methodology was well developed, but some changes are necessary:

The SHV-1 and TEM-1 genes are not considered ESBL but rather cephalosporinases, as they do not confer resistance to third and fourth generation cephalosporins, therefore, they need to be removed or replace the term ESBL with cephalosporinases where these 2 genes appear.

According to the results, there are 57 samples positive for the CTX-M gene, of which 6/57, 4/57... the respective CTX-M were identified, in addition to missing mention of the TEM-1 gene. And not "Among the ESBL-positive samples, while 76.2% (45/59) presented the CTX-M beta-lactamase, 1.6% (1/59) was positive for the SHV-1. From the samples with CTX-M, 10.1% (6/59) were CTX-M-15 and 6.7% (4/59) CTX-M-8. 3.3% (2/59) did not have the gene identified through the primers used."

Resistance to CRO, CTX, CTF and CPM must be reported separately, because although the bacterium is considered an ESBL, resistance to cephalosporins is individual and cannot be extravagant for all

Reviewer 2 Report

Comments and Suggestions for Authors

This is an interesting paper which builds on the growing one health literature around antimicrobial resistance. The study appears to be well executed and reported with a few minor inaccuracies which are detailed below. I have tried to suggest an alternative for where there are grammatical issues to aid the authors

Line 22- presence of one type of antibacterial resistance … ( I think this is what you mean, but please check and modify as it doesn’t sound correct at the moment)

Line 23- perhaps having 59/387 as a % in here maybe useful?

Line 32- stray bold letter in here

Line 34, line 130, Line 185- here is where you need to be a bit tighter- bird cloacal swabs were taken, but mammals don’t have one. Perhaps needs more details in here, and in the manuscript

Line 38- bacterial names need to be in italics

Line 42- these two sentences need to be linked as they don’t make sense as they are

Line 71- perhaps bacterial drug resistance? Or antimicrobial resistance may work better to start this sentence?

Line 84- AMR bacteria maybe better in here than AMRs

Line 139- it would be nice to have the concentrations in here for ease of reading without referring to the CLSI

Line 149- multi drug resistant bacteria …. (reword)

Table 1 is almost not needed as it is in the text. You could keep this in and combine the types and concentrations of the antibiotics in here if you prefer?

Line 155- supplementary table is a typo

Line 204- 5.9% prevalence. The other municipalities had a prevalence lower than ….. (reword)

Table 2 is a large table, is it possible to split this for ease of reading?

Line 212- being male represented a protective ….. (Reword)

Figure 2- consider the colours here as when printed this will all look the same

Line 257- it would be nice to include the number of different species of birds tested in the methodology especially as you draw conclusions from it here

Figure 4- are the confidence intervals shown? It may also be nice to include the common names of the animals in here too

Line 332- should these be full stops or commas for the population? If its full stops does it need some units to quantify it?

Line 340- presented a higher prevalence of … (reword)

Line 341- pregnancy as prophylaxis …. (reword)

Line 342- is this udder rather than under?

Line 357- one sample from a great kiskadee…. (reword)

Line 356- presence of ESBLs in wild birds []. The ESBLs were detected in one non-migratory  (reword)

Line 359- all collected in the northeast of Brazil … (reword)

Line 368- from the B lactam class …(Reword)

Line 374- in an anthropized place rather than … (Reword)

Line 382- of viable antimicrobials for the treatment of diseases …(reword)

Line 384- hydrolysed by ESBLs … (reword)

Line 410- and help limit the purchase …(reword)

Comments on the Quality of English Language

These are detailed above

Reviewer 3 Report

Comments and Suggestions for Authors

General comments

The authors have written a good manuscript on an important issue of public health concern, i. e. AMR.

The selection criteria of the study area are well-designed to capture the AMR problem with high human-animal density and the potential risk of the spread of antimicrobial genes (ARG) in free-living animals among migratory birds. The sentinel surveillance of ARG and proxy indicator of environmental contamination with antimicrobial residues have been highlighted. Socioeconomic aspects such as poverty, lack of hygiene and sanitation, literacy, and ignorance among farmers have been mentioned. The study raised the implementation of a ban on antibiotics as growth promoters in Brazil. The methodology is well described and the sampling and lab. investigation designs have been clarified. The statistical analysis has been well described. The limitation of the study has been recognized.

Specific comments

Over-the-counter sale of human and veterinary medicines is a major problem that triggers misuse and abuse of antibiotics in human treatment and animal production sectors. Poultry and pig industries consume the highest volume of antimicrobial substances worldwide and it will be good to reflect it in the Brazilian context. I noticed that a higher prevalence of AMR bacteria without treatment history has been mentioned in pigs only in the manuscript.

The authors are advised to look at published papers of similar studies such as the detection of ARG in migratory birds in Bangladesh and other countries. The citation of other author’s work should be mentioned appropriately.  The presence of antimicrobial resistance has already been reported 900 km away from 349 Piaui, in 32 individuals of semipalmated sandpiper by [49]. Please elaborate who! Silva, C.R.

Specific recommendations have been made based on study results. The One Health approach has been mentioned but it needs to be elaborated for containment of AMR in the Brazilian context. The integrated surveillance of AMR at the human-animal-environment interfaces such as ESBL e. coli tricycle surveillance should be introduced to bring health and agriculture sectors together.

130 faecal (not cloacal) samples were obtained

185 cloacal and rectal samples

Figure 3: Please mention the ‘isolated from animals' buffaloes, horses, goats, pig, poultry, and migratory bird’s common name in the graph.

Comments on the Quality of English Language

The English language needs improvement as mentioned above. 

Round 2

Reviewer 2 Report

Comments and Suggestions for Authors

I wish to thank the authors for addressing my comments